# Beyond the Known: Ambiguity-Aware Multi-view Learning

## ABSTRACT

The inherent variability and unpredictability in open multi-view learning scenarios infuse considerable ambiguity into the learning and decision-making processes of predictors. This demands that predictors not only recognize familiar patterns but also adaptively interpret unknown ones out of training scope. To address this challenge, we propose an Ambiguity-Aware Multi-view Learning Framework, which integrates four synergistic modules into an end-to-end framework to achieve generalizability and reliability beyond the known. By introducing the mixed samples to broaden the learning sample space, accompanied by corresponding soft labels to encapsulate their inherent uncertainty, the proposed method adapts to the distribution of potentially unknown samples in advance. Furthermore, an instance-level sparse inference is implemented to learn sparse approximated points in the multiple view embedding space, and individual view representations are gated by view-level confidence mappings. Finally, a multi-view consistent representation is obtained by dynamically assigning weights based on the degree of cluster-level dispersion. Extensive experiments demonstrate that our approach is effective and stable compared with other state-of-the-art methods in open-world recognition situations.

## CCS CONCEPTS

• **Computing methodologies** → **Artificial intelligence**; **Neural networks**; **Supervised learning**.

## KEYWORDS

Multi-view Learning, Open-set Recognition.

## 1 INTRODUCTION

Different feature extractors or sensors can perceive various patterns of information conveyed by real-world objects. Multi-view data provides a wealth of information that can significantly enhance learning and understanding of models in analysis tasks [7, 26]. Therefore, multi-view learning can improve the robustness and reliability of recognition performance by integrating multiple perspectives or patterns of data as compared to single-pattern information processing. Indeed, due to constraints imposed by device limitations or working conditions, it is often expensive or even infeasible to gather comprehensive multi-view data encompassing all categories during the training phase [13, 20]. Although multi-view learning methods are well-developed, most work has focused on improving performance within known constraints. Nonetheless, the inference

Permission to make digital or hard copies of all or part of this work for personal or classroom use is granted without fee provided that copies are not made or distributed for profit or commercial advantage and that copies bear this notice and the full citation on the first page. Copyrights for components of this work owned by others than the author(s) must be honored. Abstracting with credit is permitted. To copy otherwise, or republish, to post on servers or to redistribute to lists, requires prior specific permission and/or a fee. Request permissions from permissions@acm.org.

*ACM MM, 2024, Melbourne, Australia*

© 2024 Copyright held by the owner/author(s). Publication rights licensed to ACM.
ACM ISBN 978-x-xxxx-xxxx-x/YY/MM
https://doi.org/10.1145/nnnnnnn.nnnnnnn

process in real-world testing environments is plagued by ambiguity, an aspect frequently neglected in previous works. Ambiguity essentially captures the gap between the controlled conditions of model training and uncontrollable factors encountered when the model is deployed in real-world scenarios. To encapsulate, the presence of ambiguity in scenarios of open multi-view learning remains a paramount obstacle, originating from the inherent variability and unpredictability of external factors.

This ambiguity in multi-view data arises from its inherent variability, presenting a persistent issue for multi-view learning: **how to reconcile disparate views to distill essential patterns amidst the noise and redundancy.** Variability of multi-view data introduces consistency and complementarity nature that contribute to favorable inference performance [27, 29], while additional messages that tend to disrupt decision-making are also conflated. Each view may capture distinct facets of the data, leading to varied representations that necessitate reconciliation for a comprehensive comprehension. A primary concern is the redundancy brought by an excessive amount of information, which includes irrelevant details that can obscure the essential patterns critical for precise classification. Concurrently, the disparity in feature spaces across different views, known as the heterogeneity gap, complicates the amalgamation of data samples from these disparate sources [12]. To tackle these challenges, recent studies have concentrated on developing a unified representation space that integrates intrinsic information from various views, thus facilitating a more coherent interpretation of multi-view data. Additionally, by learning to identify and prioritize the most task-relevant features across views, it is possible to significantly mitigate the negative influence of redundancy.

Whereas external ambiguity introduced by unknown classes blurs the boundaries between categories in the representation space, thereby presenting issue: **How to inform multi-view classifiers about the existence of its unknown classes so that it recognizes them with low confidence.** Moreover, the open-world problem introduces safety risks, particularly when classifiers erroneously assign high confidence levels to unknown classes, potentially compromising decision performance [5]. The ability to simultaneously recognize known categories and reject unknown categories is the task of open-set recognition approaches [11, 18, 28]. Unfortunately, these methods predominantly focus on analyzing single-view feature data, which is prone to external disturbances, potentially leading to misrecognition. In contrast, multi-view learning leverages the complementary nature of diverse views to enhance the efficacy of open-set recognition. In order to avoid the problem of ambiguous category boundaries caused by insufficient categories of training multi-view data, we try to extract category information from the existing raw data for the model to adapt to the distribution of potentially unknown categories in advance. This preparation allows the model to adjust its recognition boundaries more dynamically, enhancing its ability to differentiate between known and unknown categories effectively.

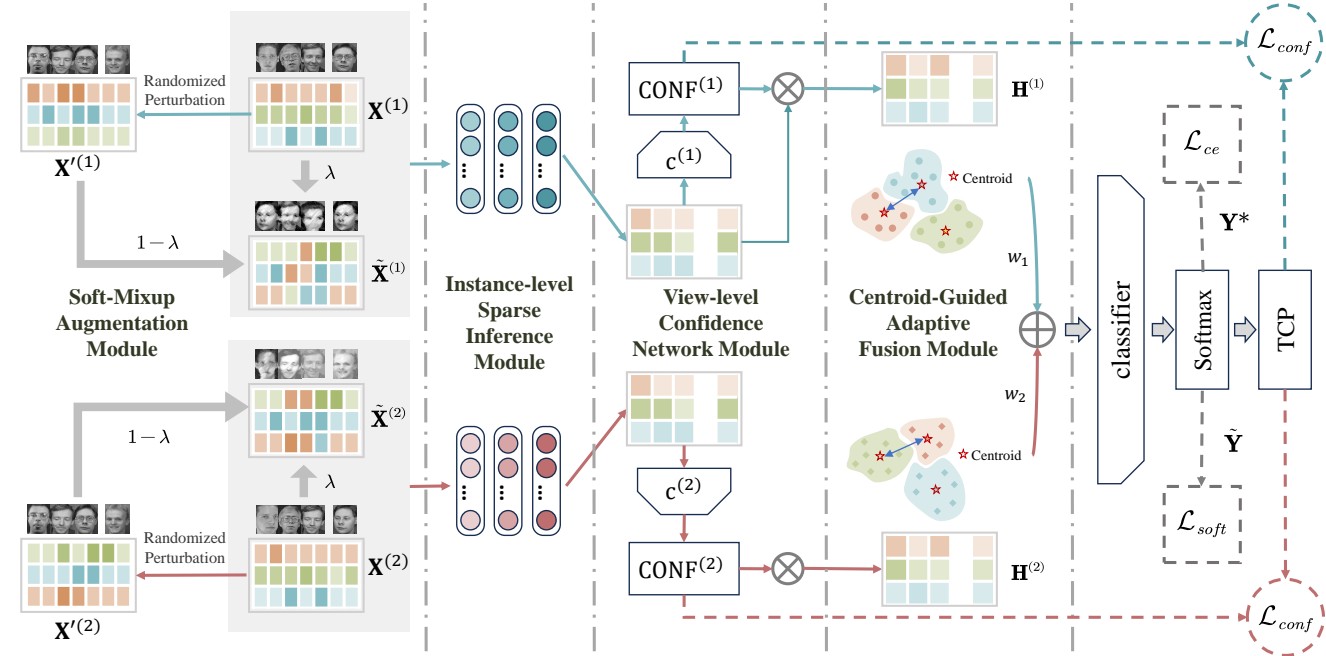

**Figure 1: Overall of the proposed Ambiguity-Aware Multi-view Learning Framework. The soft-mixup augmentation module generates mixing samples from the original ones, feeding them into the subsequent network alongside the original. The instance-level sparse inference module computes view-specific representations, while the view-level confidence network module calculates confidence maps for gating. Ultimately, a consistent representation is fused using inter-class dispersion weights via the centroid-guided adaptive fusion module. The entire framework is optimized with three losses.**

To jointly address the above-mentioned issues, in this work, we focus on generalization in open multi-view environments, aiming at exploring data distributions and underlying properties from observational data to enhance the ability to recognize potentially unknown classes. Specifically, we devise four distinct modules, each geared towards achieving specific objectives. The soft-mixup augmentation module is designed to generate soft-labelled synthetic instances in order to better manage open space risk. While the instance-level sparse inference module based on interpretable optimization objectives is applied to filter out the influence of redundant features and projects multi-view features into common representation space. Additionally, the view-level confidence network module estimates prediction confidence to gating view-specific representations. Finally, the centroid-guided adaptive fusion module dynamically modulates the contributions of different views based on inter-class dispersion, guaranteeing that the intrinsic information of multiple views is encoded into the multi-view consistent representation. The overall framework is illustrated in Fig. 1. The main contributions of this paper can be listed as follow:

- We propose Ambiguity-Aware Multi-view Learning Framework (AAML), which addresses the variability and unpredictability in open multi-view learning environments.
- We employ mixed interpolated samples to occupy the unknown ambiguous space between categories to broaden the learnable feature space, in which the soft labeling strategy measures the uncertainty.

- Extensive comparative experiments prove that the proposed AAML achieves stable correct classification rates at varying levels of false positive rate, highlighting its robustness and reliability in open-set classification tasks.

## 2 RELATED WORK

### 2.1 Multi-view Learning

Existing multi-view learning methods are usually based on the assumption that the different feature views are projected from the potential feature space. Assuming that the training multi-view dataset $\mathcal{D}_{train} = \{\{\mathbf{X}^{(v)}\}_{v=1}^{V}, \mathbf{y}^*\}$ consists of $N$ instances across $K$ known categories with $V$ views, where $\mathbf{X}^{(v)} = \{\mathbf{x}_i^{(v)}\}_{i=1}^{N} \in \mathbb{R}^{N \times D^{(v)}}$ and corresponding true labels $\mathbf{y}^* = \{y_i^*\}_{i=1}^{N}$ with $y^* \in \mathcal{Y} = [1, K]$. In a multi-view scenario, the first step is to construct view-specific mappings to project various feature spaces $\mathcal{X}^{(v)}$ to uniform space $\mathcal{H}$, where $m^{(v)} : \mathcal{X}^{(v)} \rightarrow \mathcal{H}$, where $d^{(v)}$ represents the dimensional of original features in $v$-th view.

Such approaches adhere to a specific-uniform procedure where potential features are first extracted from each view and then combined to create a uniform representation. The main goal is to take advantage of the complementary and consistent information provided by multiple views, and to reduce the interference of invalid information such as redundancy and noise. To avoid degrading the quality of the derived representation, Wang *et al.* [19] proposed

MetaViewer to guide the learning of uniform representations, formulating the extraction and fusion of view-specific features as a nested optimization problem. Similarly, Tang *et al.* [17] proposed deep network to achieve multi-view safeness by automatically selecting features while extracting complementary information and eliminating irrelevant noise. In order to keep the views with clear clustering structures from receiving constraints from views with ambiguous structures, Chen *et al.* [2] utilized global discriminative information to guide the learning of local common representation. Tang *et al.* [16] projected each view into a label space with consensus part and view-specific part, while cross-view similarity graph learning term is embedded to preserve the local structure. Liang *et al.* [14] factorized task-relevant information in multi-view data into shared and unique information, and removed task-irrelevant information via upper bounds on mutual information. Despite the increasing interest in multi-view learning methods, rarely has research focused on tailoring these approaches to open settings.

## 2.2 Open-set Recognition

Open-set recognition addresses confronts the challenge of correctly classifying samples from known categories while accurately identifying those from unknown categories. The main challenge is that incomplete knowledge exists in the training phase and potential unknown categories can be encountered in an algorithm during testing [6, 24]. Following predictive distribution $P(y \mid \mathbf{x}_i; \Theta)$ calculated by the classifier $f : \mathcal{H} \rightarrow \mathcal{Y}$, the entire classification network can be trained by using loss functions such as cross-entropy loss, defined as

$$\mathcal{L}_{ce} = - \sum_{(\mathbf{x}_i, y_i^*) \in \mathcal{D}_{train}} \left( \mathbb{I}^{y_i^*} \log \mathbf{p}(y \mid \mathbf{x}_i; \Theta) \right) \quad (1)$$

where $\mathbb{I}^{y_i^*}$ denotes the one-hot label vectors, and $\Theta$ is a set of model parameters. However, due to the closed-world property $\sum_{i=1}^{K} P(i \mid \mathbf{x}_i; \Theta) = 1$, it can mistakenly classify novel class instances with high confidence. To identify instances from outside the predefined category set, a straightforward strategy is to set thresholds on the predicted prediction, the instance is conservatively labeled as "unknown" if the score is below the threshold $\delta$. Thus it can be inferred that the class predicted as

$$\hat{y}_i = \begin{cases} \arg\max_{j \in \mathcal{Y}} P(j \mid \mathbf{x}_i; \Theta), & \text{if } P(j \mid \mathbf{x}_i; \Theta) \geq \delta \\ \text{unknown class} & \text{otherwise.} \end{cases} \quad (2)$$

However, simply applying manually defined thresholds to all known categories may not always be applicable or effective, especially when dealing with multi-view data with various representation spaces. Such an approach ignores the inherent complexity and diversity among different categories as well as among different views, which may lead to misclassification and poor recognition of novel categories. It's essential to consider a more tailored strategy that respects the complementarity and consistency nature of multi-view datasets to ensure robust classification and effective identification of new category instances beyond the known.

## 3 THE PROPOSED FRAMEWORK

The proposed model consists of four key components: 1) **Soft-Mixup Augmentation Module** introduces a soft labeling strategy

for these synthetic interpolated data to measure their uncertainty. 2) **Instance-level Sparse Inference Module** employs a data-driven feedforward network to learn sparse approximated points in the multiple view embedding space; 3) **View-level Confidence Network Module** develops the view-level confidence map to approximate true class probability for the trustworthiness of predictions; 4) **Centroid-Guided Adaptive Fusion Module** adopts the interclass dispersion to metric the informativeness of each view for reliable fusion.

*3.0.1 Soft-Mixup Augmentation Module.* Effective classifiers must distinguish known classes and adeptly manage the risk associated with "open space". This entails avoiding the overextension of class boundaries without cannibalizing the uncharted areas of the feature space that could belong to unrecognized categories. Drawing from the Mixup approach [25], we apply linear interpolation on two samples drawn at random from the training data to produce additional virtual outliers. A random parameter $\lambda \in [0, 1]$ is sampled as $\lambda \sim \mathbf{Beta}(\alpha, \alpha)$. We select $\mathbf{x}_i^{(v)}$ and $\mathbf{x}_j^{(v)}$ in the $v$-th view to generate the augmented samples $\mathcal{D}_{mix} = \{\{\tilde{\mathbf{X}}^{(v)}\}_{v=1}^V, \tilde{\mathbf{y}}\}$ to enhance the generalization ability by familiarizing it with the open space near existing class clusters. For each view, we perform a randomized perturbation to selection $\mathbf{x}'^{(v)}$, which can be obtained as follows

$$\tilde{\mathbf{x}}^{(v)} = \lambda \mathbf{x}_i^{(v)} + (1 - \lambda) \mathbf{x}'^{(v)}_j, \quad (3)$$

where $\lambda$ determines the contribution of each original sample to features of the mixed samples. Since the samples produced by linear interpolation may lie in the neighborhood of another known class, simply assigning them to the unknown category may confuse the judgement of the predictor. Therefore, we employ a soft label $\mathbb{I}^{\tilde{y}}$ to teach the model to be uncertain for artificially created instances as

$$\mathbb{I}^{\tilde{y}} = \lambda \mathbb{I}^{y_i} + (1 - \lambda) \frac{1}{|\mathcal{Y}|} \sum_{y_j \in \mathcal{Y}} \mathbb{I}^{y_j}. \quad (4)$$

Soft labels of mixed samples weight the distribution of class-specific metrics and all classes for an original sample, reflecting the probability that the sample belongs to each possible class. At the same time, by ensuring that unknown samples have a uniform probability distribution across known categories, the model avoids making overconfident predictions about never before learned categories. This approach mitigates potential classifier confusion by not strictly assigning these mix samples to an existing class but instead indicating their intermediate nature. Achieving a maximum entropy distribution across a set of categories indicates maximum uncertainty about which category a sample belongs to, implying that the model assigns probabilities across all categories as evenly as possible for unknown samples. Thus the learning objective for mixed samples is defined as

$$\mathcal{L}_{soft} = - \sum_{(\tilde{\mathbf{x}}_i, \tilde{y}_i) \in \mathcal{D}_{mix}} \left( \left( \mathbb{I}^{\tilde{y}_i} \log \mathbf{p}(y \mid \tilde{\mathbf{x}}_i; \Theta) \right) \right). \quad (5)$$

Upon generating these mixed instances, the entire dataset represented as $\mathbf{X}^{(v)} = \{\mathbf{X}^{(v)}, \tilde{\mathbf{X}}^{(v)}\}$, is fed into the proposed network. This process involves the integration of original and augmented data, enriching the learning context and bolstering its capability to discern and represent complex data patterns, ultimately leading

to improved task performance, especially in scenarios involving categories not present in the training data.

*3.0.2 Instance-level Sparse Inference Module.* In order to strive balance between model complexity and learning capacity, we design a data-driven feedforward network. To learn compact feature representations $\mathbf{h} \in \mathbb{R}^K$ of multi-view samples, the function involves data fitting terms term and certain regularizations, expressed as

$$\mathbf{h} = \arg \min_{\mathbf{h}} \left\{ \lambda \mathcal{R}(\mathbf{h}) + \sum_{v=1}^{V} w_v \mathcal{F}(\mathbf{h}, \mathbf{x}) \right\}. \quad (6)$$

where $\lambda$ denotes a regularization parameter, and $\mathbf{w} = [w_i]_{i=1}^{V}$ denotes a weight vector for balancing the contributions from distinct views. This formulation encapsulates the integration of data-fitting terms and various regularizations, aiming to implement efficient category separation in low-dimensional spaces.

Specify, we first project diverse original feature spaces into unified dimensional space in preparation for subsequent fusion. To address the challenges inherent in high-dimensional problems, we turn our focus to the concept of model sparsity. Herein, we invoke the sparsity of the combined $\ell_1$-norm to suppress irrelevant features. Designating $\Phi$ as the view-specific projection matrix. Since the objective is a quadratic optimization with a non-differentiable regularizer problem. Formally, this corresponds to the case where the proximal operator defined for a convex regularizer, given by

$$\mathbf{prox}_{\lambda\mathcal{R}}(\mathbf{h}) = \arg \min_{\mathbf{u}} \lambda \mathcal{R}(\mathbf{u}) + \frac{1}{2}\|\mathbf{u} - \mathbf{h}\|_2^2. \quad (7)$$

At each iteration of the proximal algorithm, the current value of $\mathbf{h}$ is updated as the solution of the proximal problem

$$\begin{aligned} \mathbf{h}_{t+1} &= \mathbf{prox}_{\lambda\mathcal{R}} \left( \mathbf{h}_t - \frac{\lambda}{L_f} \mathrm{grad}_{\mathbf{h}} \mathcal{F}(\mathbf{h}_t) \right), \\ &= \mathbf{prox}_{\lambda\mathcal{R}} \left( \mathbf{h}_t - \frac{\lambda}{L_f} (\mathbf{h}_t \Phi - \mathbf{x}) \Phi^T \right), \end{aligned} \quad (8)$$

where $L_f$ is the Lipschitz constant of $\mathcal{F}$. Taking inspiration from the Learned Iterative Shrinkage and Thresholding Algorithm network [8], the optimization strategy can be effectively unrolled into a sequence of updates within a feedforward network. Here, the components that need to be precomputed are parameterized as a fully connected layer within the network, enabling it to learn adaptively from the data. To specifically address different views in the data, we extend our network layer update formulation as follows

$$\mathbf{h}_{t+1}^{(v)} = \sigma_\tau \left( \mathbf{h}_t^{(v)} \mathbf{U}_t^{(v)} + \mathbf{x}^{(v)} \mathbf{V}_t^{(v)} \right) \quad (9)$$

where $\mathbf{U}_t^{(v)} = \mathbf{I} - \Phi\Phi^T$ and $\mathbf{V}_t^{(v)} = \Phi^T$ serve to refine the view-specific transformations at each iteration. And the update formula is defined as $\sigma_\tau(\mathbf{x}) = \mathbf{F}(\mathbf{x} - \tau) - \mathbf{F}(-\mathbf{x} - \tau)$, where $\tau$ represents a learnable threshold parameter, and $\mathbf{F}(\cdot)$ denotes an activation function such as ReLU, SELU, ELU, etc. The above formulation positions each network layer to correspond to an optimization algorithm iteration, enhancing the learning process of traditional optimization models through parameterized updates. Besides, we adopt a one-step inference, simplifying the process by retaining a single iteration of this optimization, thus promoting a balance between efficiency and performance.

This module adopts a straightforward mathematical model inspired by sparse coding principles to construct the entire network structure. This choice is driven by two primary objectives: 1) To

ensure that a sufficiently discriminative sparse representation is learned from both the original and mixed-sample features, reducing the interference of redundant information for improved adaptation to the open world. 2) To improve interpretability, with simpler models offering clearer insights into the dynamics between responses and covariates.

*3.0.3 View-level Confidence Network Module.* Adopting the Maximum Softmax Probability $\mathrm{MCP} = P(y = \hat{y} \mid \mathbf{x}; \Theta)$ as the confidence score usually leads to over-confidence, particularly in cases of incorrect predictions [3]. To counteract this issue, we consider the True Class Probability (TCP) as the target confidence value, defined as the predicted probability of the ground truth class $\mathrm{TCP}(\mathbf{x}, y^*) = P(y = y^* \mid \mathbf{x}; \Theta)$. To obtain a confidence score estimate for each view, an auxiliary view-specific confidence network module $c^{(v)}$ parameterized by $\theta_c$ is customized as

$$\mathrm{CONF}^{(v)} = c^{(v)} \left( \mathbf{x}^{(v)}; \theta_c \right), \quad (10)$$

where the sigmoid activate function is employed after the confidence network to ensure that $\mathrm{CONF}^{(v)}$ is normalized within the $[0, 1]$. $\mathrm{CONF}^{(v)}$ denotes a confidence map for each view, the goal of this module is to train it such that $\mathrm{CONF}^{(v)}$ closely approximates the TCP, utilizing a mean-square-error loss for this purpose

$$\mathcal{L}_{conf} = \sum_{v=1}^{V} \sum_{i=1}^{N} \left( c^{(v)} \left( \mathbf{x}_i^{(v)}; \theta_c \right) - \mathrm{TCP}(\mathbf{x}_i, y^*) \right)^2. \quad (11)$$

To further refine the capacity to leverage view-specific information effectively, a view-level gating strategy is employed as

$$\mathbf{H}^{(v)} = \mathrm{CONF}^{(v)} \mathbf{H}^{(v)}. \quad (12)$$

This approach ensures that the learned view-specific representations are informed by the reliability of predictions, thereby enhancing the overall predictive performance and trustworthiness.

*3.0.4 Centroid-Guided Adaptive Fusion Module.* Recognizing the variability in quality across different views, where less informative views may dilute overall performance, we strategically assign weights to each view. Essentially, by emphasizing more informative views and diminishing the impact of less informative ones, we can achieve a more effective and reliable integration.

Intuitively, the proximity of centroids within views is negatively correlated with the discriminative capability of each single view embedding. To quantify this, we first identify the centroids $\mathbf{o}_i^{(v)}$ of known category within the $v$-th view embedding space as below

$$\mathbf{o}_i^{(v)} = \frac{1}{|C_i|} \sum_{j: \mathbf{y}_j = i} \mathbf{h}_j^{(v)}, \mathbf{h}_j^{(v)} \in \mathbf{H}^{(v)}, \quad (13)$$

where $|C_i|$ is the number of training instances in category $C_i$. Subsequently, we calculate the inter-category distance from all centroids of each view. To avoid making the largest distance between two classes overly influence the measure of interclass variation, we focus on the minimum distance between any two categories as a metric for view discriminativeness, denoted as

$$d_v = \min \left\{ Dist \left( \mathbf{o}_i^{(v)}, \mathbf{o}_j^{(v)} \right) \right\}, i, j \in \mathcal{Y} \text{ and } i \neq j, \quad (14)$$

where $Dist(\cdot, \cdot)$ is the distance function, which adopts the Euclidean distance in this work. This strategy provides a balanced assessment of the contribution of the different views. Building on this

**Algorithm 1** Ambiguity-Aware Multi-view Learning Framework

---

**Require:** Multi-view data $\mathcal{D}_{train}$, training epoch $e$, regularization parameters $\alpha, \beta$, learning rate $\eta$.

**Ensure:** Optimized model parameters $\Theta$.

1: Initialize $\{\mathbf{U}^{(v)}, \mathbf{V}^{(v)}\}_{v=1}^{V}$;
2: **for** $i = 1 \rightarrow e$ **do**
3:    **for** $v = 1 \rightarrow V$ **do**
4:       Genrate the augmented samples $\{\tilde{\mathbf{X}}^{(v)} \in \mathbb{R}^{m \times D^{(v)}}, \tilde{\mathbf{y}}\}$ for joint training as Eq. (4);
5:       Calculate view-specify representation $\mathbf{H}^{(v)}$ as Eq. (9);
6:       Compute confidence map $\mathrm{CONF}^{(v)}$ though view-specific confidence network module as Eq. (10);
7:       Update $\mathbf{H}^{(v)}$ with view-level gating $\mathrm{CONF}^{(v)}$ as Eq. (12);
8:       Calculate the weight of multiple views $w^{(v)}$ as Eq. (15);
9:    **end for**
10:   Fusing multiple view representations to obtain view-consistent representation $\mathbf{H}$ as Eq. (16);
11:   Obtain predictive distribution $P(y \mid \mathbf{x}_i; \Theta)$ by classifier $f$;
12:   Compute the loss of joint optimization as Eq. (17);
13:   Update the parameters $\Theta$ though backward propagation;
14: **end for**
15: **return** Optimized $\Theta$.

---

foundation, we perform class-wise $\ell_1$-norm on the reciprocal distances: $\hat{\mathbf{d}} = \mathbf{d}^{-1} / \left\| \mathbf{d}^{-1} \right\|_1$. By normalizing the reciprocal of these distances, views with more distinctive category separations are afforded greater weights. Accordingly, the weight assignments $\mathbf{w}$ can be determined as

$$w_v = \frac{\exp\left(-\hat{d}_v\right)}{\sum_{j=1}^{V} \exp\left(-\hat{d}_j\right)}, \quad \sum_{v=1}^{V} w_v = 1. \tag{15}$$

Ultimately, these weights are employed to construct the multi-view consistent representation $\mathbf{H}$ through a weighted summation of individual view embeddings

$$\mathbf{H} = \sum_{v=1}^{V} w_v \mathbf{H}^{(v)} \tag{16}$$

Through this method to achieve balanced and insightful fusion of multiple views, this embedding is expected to be more discriminative than any single-view setting.

### 3.1 Overall Training Objective

Additional classifier, denoted as $f : \mathcal{H} \rightarrow \mathcal{Y}$, is tailored to process view-consistent representation $\mathbf{H}$. The output from the final fully-connected layer is subsequently processed through the SoftMax function, generating a probability distribution over the labels of all $K$ recognized classes. The overall learning objective of the model is encapsulated by the following composite loss function

$$\mathcal{L} = \mathcal{L}_{ce} + \alpha \mathcal{L}_{conf} + \beta \mathcal{L}_{soft}. \tag{17}$$

$\mathcal{L}_{ce}$ plays a fundamental role in guiding the model toward the accurate classification of known categories on the training data. $\mathcal{L}_{conf}$ facilitates the alignment of the output of the confidence network with the true confidence level as indicated by TCP, thereby enhancing the reliability, particularly in its confidence estimations. $\mathcal{L}_{soft}$

Table 1: A brief description of the tested multi-view datasets.

| Datasets | Samples | Dimension of Views | Classes |
|---|---|---|---|
| ESP-Game | 11,032 | 100/100 | 7 |
| Flower17 | 1,360 | 1,360/1,360/1,360/1,360/1,360/1,360/1,360 | 17 |
| MNIST | 2,000 | 1,930/9/30 | 10 |
| Reuters | 1,500 | 21,531/24,892/34,251/15,506/11,547 | 6 |
| ORL | 400 | 512/59/864/254 | 40 |
| Youtube | 2,000 | 2,000/1,024/64/512/64/647 | 10 |

is crafted to minimize prediction error on these mixed instances, enhancing its ability to generalize from known to unknown categories. This objective not only enhances confidence accuracy but also improves adaptability to unknown categories, ensuring robust and versatile performance in various real-world applications. The entire procedure is outlined in Algorithm 1.

The core computations for computing multi-view consistent representation are 1) creation of $M$ mixed samples where linear interpolation is $O(MD^{(v)})$; 2) one-step sparse inference, for which the complexity is $O((N+M)H^2 + (N+M)(D^{(v)})^2)$, accounting for operations on both the original $N$ samples and the $M$ mixed samples.; 3) the gating strategy is $O((N+M)H)$, and 4) Computing the centroids for each category has a complexity of $O(\sum_{i=1}^{K} |C_i| D^{(v)}) = O(ND^{(v)})$ and calculating inter-category distances between $K(K+1)/2$ pairs of centroids is $O(K^2 D^{(v)}/2)$. Given that $N$ is typically larger than $M$, the overall time complexity becomes $O(VNH^2 + \sum_{v=1}^{V}(N(D^{(v)})^2) + K^2 D^{(v)}))$ under multi-view setting for each training epoch.

## 4 EXPERIMENTAL RESULTS AND STUDY

### 4.1 Benchmark Datasets

Our experiments are conducted on six well-known multi-view datasets. The statistics of these datasets are summarized in Table 1, with detailed descriptions provided below.

1) **ESP-Game**[1] is a social image collection searched from an image annotation website where two players without any communication try to predict the same tags for a test image. 2) **Flower17**[2] consists of different flower categories that are common in the UK. The images are available in a variety of views, scales and photometric variations. 3) **MNIST**[3] is a well-known dataset of handwritten digits with IsoProjection, LDA, and NPE features. 4) **ORL**[4] is a face image dataset, taken at varying lighting, different times, facial expressions, and facial details. 5) **Reuters**[5] consists of multilingual document corpora in five languages: English, French, German, Spanish, and Italian. 6) **Youtube**[6] is a video game dataset including both visual (Cuboids Histogram, Histogram of Motion Estimate, and HOG features) and audio features (MFCC, Volume Streams, and Spectrogram Streams).

---

[1] https://www.cs.cmu.edu/~biglou/resources/
[2] http://www.robots.ox.ac.uk/vgg/data/flowers/
[3] http://yann.lecun.com/exdb/mnist/
[4] http://cam-orl.co.uk/facedatabase.html
[5] https://archive.ics.uci.edu/dataset/259/reuters+rcv1+rcv2+multilingual+multiview+text+categorization+test+collection
[6] https://archive.ics.uci.edu/dataset/269/youtube+multiview+video+games+dataset

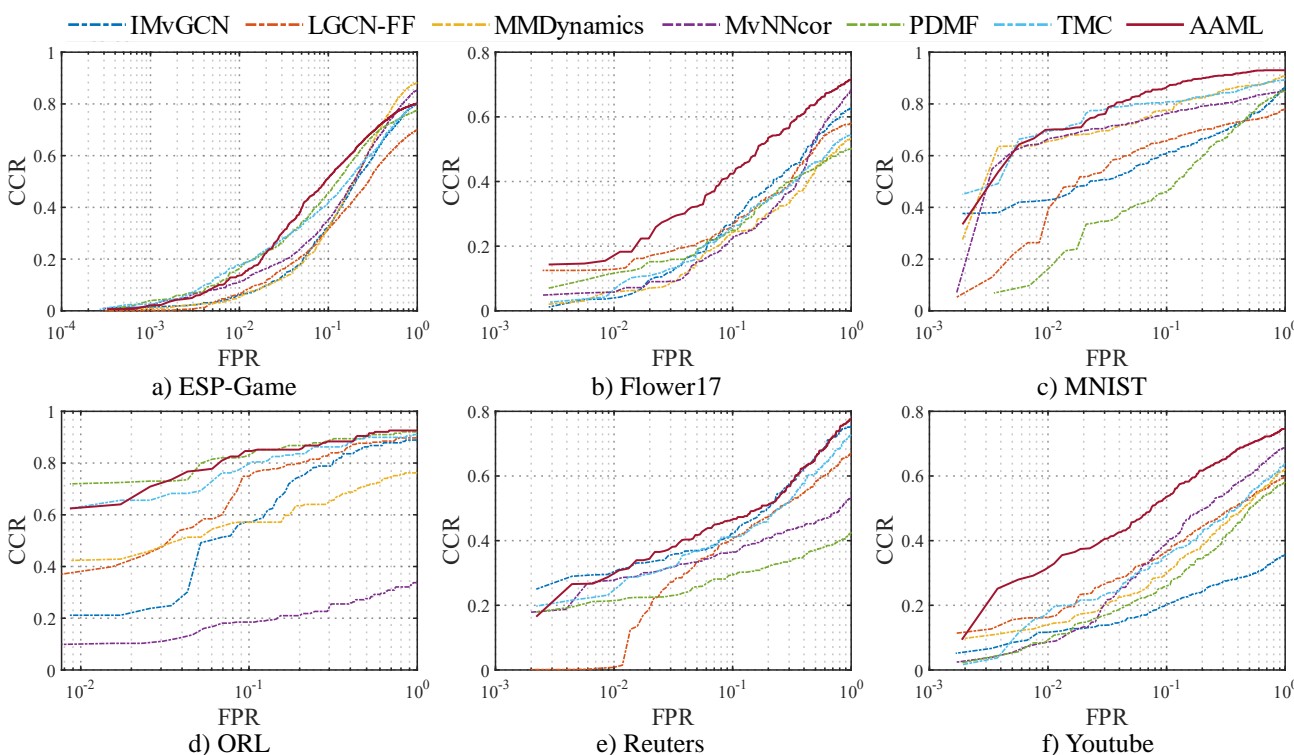

Figure 2: OSCR curves plotting the CCR over the FPR on all test multi-view datasets for all compared methods.

## 4.2 Compared Methods

We compare the proposed method with several state-of-the-art methods for multi-view representation learning, including

1) **IMvGCN** [21] combines the reconstruction error and Laplacian embedding, accompanied by a differentiable orthogonal normalization constrain to improve generalization capability. 2) **LGCN-FF** [1] integrates sparse autoencoders with a learnable GCN, enabling the simultaneous extraction of feature representations and node relationships within graphs. 3) **MMDynamics** [9] models both the feature-level and modality-level informativeness for trustworthy fusion. 4) **MvNNcor** [23] models view-specific information and cross-correlations information through an interactive network to jointly make decisions and infer categories. 5) **PDMF** [22] earns relations and the auxiliary representation through pre-training to tune the mappings from the original data to the comprehensive representation. 6) **TMC** [10] conducts decision fusion according to uncertainty estimation of multiple views.

## 4.3 Evaluation Metric

To handle known and unknown samples separately, we introduce the Open-Set Classification Rate (OSCR) [4]. By setting a probability threshold $\theta$, it can balance sensitivity to known classes with the ability to reject unknowns. For samples from known categories, we calculate the Correct Classification Rate (CCR) as the fraction of the samples where the correct class $y^*$ has maximum probability and has a probability greater than $\theta$. Concurrently, False Positive Rate (FPR) is determined by the fraction of samples from the unknown

category that are classified as any known class $y \in \mathcal{Y}$ with a probability greater than $\theta$. Different applications may tolerate different levels of FPR for the benefit of higher CCR, making these specific metrics highly relevant for tuning the models according to specific needs. For security-critical applications such as biometrics or fraud detection, a low FPR is prioritized over CCR to block unauthorized access. Conversely, content recommendation or advertising systems may allow a higher FPR to enhance inclusivity and user experience by ensuring relevant content is not overlooked. In order to measure the quality of representation, we evaluate model performance with **CCR at FPR of 0.5%/1.0%/5.0%/10.0%**. Regarding the choice of thresholds, which is randomized, we follow the experimental setup in [4] by taking the maximum probability set of predicted unknown samples and traversing it to compute the CCR and FPR.

## 4.4 Implementation Details

In our experimental setting, we use the concept of **openness** [15],

$$openness = 1 - \sqrt{\frac{2 \times C_{train}}{C_{train} + C_{test}}}, \quad (18)$$

where $C_{train}$ is the number of known classes during training, and $C_{test}$ is the total number of known and unknown classes during testing. The proposed AAML has been implemented using the PyTorch framework on one NVIDIA Geforce RTX 4080 with GPU of 16GB memory. In practical application, rather than pre-generating all mixing samples, we adopt linearly transforming each mini-batch during the training process. Our training protocol involves 100 epochs across all benchmarks, maintaining a batch size of 100. In

**Table 2: CCR at different FPR are given for all compared algorithms tested on ESP-Game, Flower17 and MNIST under *openness*=0.1 setting.**

| Datasets \ Methods | ESP-Game | | | | Flower17 | | | | MNIST | | | |
|---|---|---|---|---|---|---|---|---|---|---|---|---|
| CCR at FPR of | 0.5% | 1.0% | 5.0% | 10% | 0.5% | 1.0% | 5.0% | 10% | 0.5% | 1.0% | 5.0% | 10% |
| IMvGCN | 0.0319 | 0.0601 | 0.1891 | 0.3271 | 0.0117 | 0.0365 | 0.1711 | 0.2675 | 0.3792 | 0.4271 | 0.5409 | 0.6058 |
| LGCN-FF | 0.0273 | 0.0680 | 0.2119 | 0.3169 | 0.1250 | 0.1262 | 0.2014 | 0.2616 | 0.1302 | 0.2636 | 0.6022 | 0.6559 |
| MMDynamics | 0.028 | 0.0525 | 0.1826 | 0.3223 | 0.0205 | 0.057 | 0.133 | 0.2427 | 0.6357 | 0.6527 | 0.7226 | 0.7754 |
| MvNNcor | 0.08 | 0.1109 | 0.2451 | 0.3512 | 0.0486 | 0.0567 | 0.1435 | 0.2211 | 0.5485 | 0.6448 | 0.7261 | 0.7624 |
| PDMF | **0.0977** | **0.1602** | 0.3406 | 0.4537 | 0.0702 | 0.1111 | 0.1725 | 0.2485 | 0.0687 | 0.0980 | 0.3961 | 0.4598 |
| TMC | 0.0460 | 0.0820 | 0.2750 | 0.3911 | 0.0263 | 0.0424 | 0.1579 | 0.2558 | 0.4920 | 0.6916 | 0.7924 | 0.8074 |
| AAML (w/o Soft-Mix) | 0.0265 | 0.0560 | 0.1949 | 0.3199 | 0.0965 | 0.1477 | 0.2719 | 0.3699 | **0.6208** | 0.6597 | 0.7495 | 0.8014 |
| AAML | 0.0851 | 0.1354 | **0.3950** | **0.5140** | **0.1433** | **0.1550** | **0.3216** | **0.4240** | 0.5369 | **0.6996** | **0.8174** | **0.8623** |

**Table 3: CCR at different FPR are given for all compared algorithms tested on ORL, Reuters, and Youtube under *openness*=0.1 setting. "−" indicates the out-of-memory error, and "N/A" indicates that the method does not achieve CCR at this FPR value.**

| Datasets \ Methods | ORL | | | | Reuters | | | | Youtube | | | |
|---|---|---|---|---|---|---|---|---|---|---|---|---|
| CCR at FPR of | 0.5% | 1.0% | 5.0% | 10% | 0.5% | 1.0% | 5.0% | 10% | 0.5% | 1.0% | 5.0% | 10% |
| IMvGCN | N/A | 0.2116 | 0.3016 | 0.5714 | **0.2898** | **0.2955** | 0.3693 | 0.4205 | 0.0644 | 0.0714 | 0.2032 | 0.2847 |
| LGCN-FF | N/A | 0.3704 | 0.5514 | 0.7490 | 0.0011 | 0.0079 | 0.3311 | 0.4018 | 0.1270 | 0.1611 | 0.2992 | 0.3635 |
| MMDynamics | N/A | 0.4233 | 0.5132 | 0.5714 | - | - | - | - | 0.1107 | 0.1378 | 0.2304 | 0.3018 |
| MvNNcor | N/A | 0.0988 | 0.1358 | 0.1852 | 0.1874 | 0.2772 | 0.3333 | 0.3636 | 0.0389 | 0.081 | 0.2738 | 0.3881 |
| PDMF | N/A | **0.7196** | 0.7354 | 0.8254 | 0.1946 | 0.2131 | 0.2571 | 0.2940 | 0.0443 | 0.0845 | 0.2032 | 0.2596 |
| TMC | N/A | 0.6243 | 0.6825 | 0.7884 | 0.2159 | 0.2315 | 0.3679 | 0.4119 | 0.0372 | 0.1680 | 0.2827 | 0.3561 |
| AAML (w/o Soft-Mix) | N/A | 0.3175 | 0.6349 | 0.7460 | 0.1818 | 0.2202 | 0.3452 | 0.4077 | 0.1690 | 0.2002 | 0.3793 | 0.4507 |
| AAML | N/A | 0.6243 | **0.7672** | **0.8466** | 0.2656 | 0.2869 | **0.4176** | **0.4645** | **0.2515** | **0.3099** | **0.4406** | **0.5362** |

instance-level sparse inference module, the inital value of $\tau$ is set to 0.01, and the activation function $\mathbf{F}(\cdot)$ adopts SELU. The dimensionality denoted as $h$, of the common space, is uniformly set to 32 for all datasets. The latent view-specific representations are initialized using a 1-layer fully connected layer. Besides, the trade-off parameter $\alpha$ and $\beta$ are set within $\{0.1, 1.0, 10\}$. During training, only ten percent of the labeled data are utilized for model training, with an additional ten percent reserved for validation purposes. Finally, we employ the Adam optimizer with an initial learning rate of 0.01 to optimize the AAML.

## 4.5 Experimental Results

In all the experiments below, all our open experiments have compared the proposed AAML with other classification methods under the condition of *openness*=0.1.

1) The OSCR curves of different methods on six datasets are depicted in Fig. 2. The proposed method (red solid line) overwhelmingly outperforms the other methods (colored dashed lines) in all cases, particularly notable in the Flower17 and Youtube datasets, which demonstrates its ability to maintain a high CCR while controlling the FPR. Specifically, it excels in minimizing misclassifications of unknown categories while accurately classifying known classes. This capability ensures that our model is adaptable to different applications with varying FPR requirements, making it a versatile and effective solution.

2) In order to conduct a numerical analysis, we computed the values of CCR at several representative FPR values, organized in Tables

**Table 4: Classification accuracy of all compared methods on multi-view datasets under *openness*=0 setting.**

| Datasets \ Methods | ESP-Game | Flower17 | MNIST | ORL | Reuters | Youtube |
|---|---|---|---|---|---|---|
| IMvGCN | 0.7130 | 0.4286 | 0.8352 | 0.7889 | 0.7200 | 0.3712 |
| LGCN-FF | 0.6880 | 0.6478 | **0.9196** | 0.8131 | 0.6786 | 0.5713 |
| MMDynamics | 0.7207 | 0.1961 | 0.6273 | 0.4261 | - | 0.4899 |
| MvNNcor | **0.8541** | 0.6533 | 0.8832 | 0.4506 | 0.3236 | 0.6546 |
| PDMF | 0.6665 | 0.3993 | 0.8632 | 0.6183 | 0.3094 | 0.3688 |
| TMC | 0.8195 | 0.5099 | 0.8757 | 0.8075 | 0.6778 | 0.6451 |
| AAML | 0.8402 | **0.6911** | 0.8757 | **0.8286** | **0.7406** | **0.7221** |

2 and 3, where the best performance is highlighted in bold and the second best result is underlined. We report the performance of our method without soft-mixup augmentation module as an ablation comparison. It can be observed that AAML achieves the highest performance on all datasets under CCR at FPR of 5.0% and 10.0%. While PDMF and TMC occasionally outperform AAML in specific cases, our method demonstrates more stable performance across all scenarios. Overall, AAML showcases significant advancements over existing methods in open environments, particularly in maintaining high CCR under varying FPR requirements. Furthermore, we observe that augmenting the network with synthetic data through the soft interpolation method leads to further improvements in performance compared to the network without the module. This validates our capability to capture the distribution of potentially unknown classes of data and enhances representation learning.

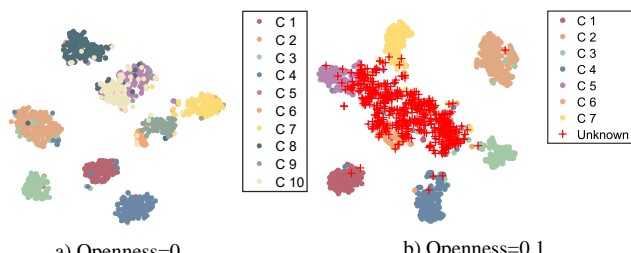

a) Openness=0                    b) Openness=0.1

**Figure 3: Visualization of representation with t-SNE learned by AAML on the MNIST dataset under different settings.**

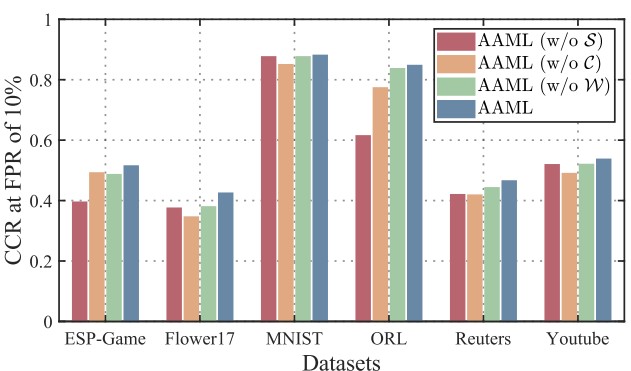

**Figure 4: Comparison among variants on all test multi-view datasets.**

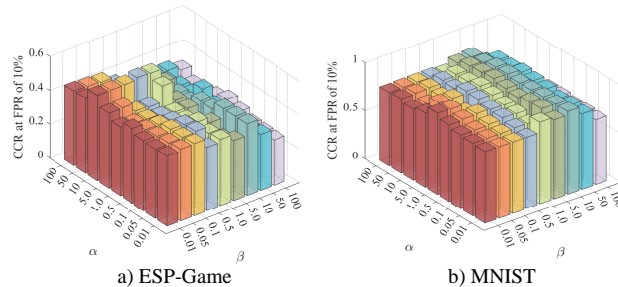

a) ESP-Game                    b) MNIST

**Figure 5: Parameter sensitivity analysis of $\alpha$ and $\beta$ in AAML on ESP-Game and MNIST datasets.**

3) In addition to the above results based on the open setting, we report the accuracy of compared methods on each dataset with the *openness* equals 0in Table 4, where the problem is completely closed. The outcomes demonstrate that the proposed model maintains competitive classification performance on known classes, affirming its capability to adeptly navigate the heterogeneity and redundancy inherent in multi-view data.

4) Fig. 3 presents scatter diagrams of AAML with different *openness* on MNIST. while colored dots signify samples from known classes, while red crosses indicate samples from unknown classes. The results intuitively highlight that the classes, as defined by the known labels, exhibit greater compactness and distinct separation. It is evident that samples belonging to the unknown class predominantly cluster around the centroid, albeit with two classes seemingly merging with it. In contrast, the other classes are situated farther away from the unknown class, emphasizing their distinctness.

## 4.6 Component and Parameter Analysis

*4.6.1 Module Analysis.* We empirically analyze the effectiveness of the other three modules. AAML (w/o $\mathcal{S}$): In this setup, we exclude the instance-level sparse inference module. This implies that we retain only the network structure within Eq. (9) without performing proximal updates to preserve sparsity. AAML (w/o $\mathcal{C}$): Here, we eliminate the view-level gating mechanism and the associated confidence loss $\mathcal{L}_{conf}$ from our framework. AAML (w/o $\mathcal{W}$): This

configuration involves directly employing the average fusion strategy, bypassing the centroid-guided adaptive fusion module. We have the following observations from Fig. 4: AAML (w/o $\mathcal{S}$) exerts the most significant impact on the ESP-Game and ORL datasets. AAML (w/o $\mathcal{C}$)demonstrates a modest decrease across all datasets, although its effect is less pronounced compared to module $\mathcal{S}$. On the other hand, AAML (w/o $\mathcal{F}$) exhibits the smallest impact, resulting in only a slight decrease in performance across all datasets. Despite the varying degrees of impact observed across these modules, our comprehensive analysis underscores the importance of AAML, as it consistently outperforms the ablated versions across all datasets. This suggests that the synergistic integration of all modules leads to the best overall performance.

*4.6.2 Parameter Analysis.* To investigate the influence of the trade-off parameter $\alpha$ and $\beta$ in Eq. (17), we conduct a sensitivity test by varying from 0.001 to 100. Fig. 5 shows the influence of different parameter values with respect to CCR at FPR at 10% on the ESP-Game and MNIST datasets. Moreover, our method performs stably when the values of $\alpha$ and $\beta$ are within a certain region.

## 5 CONCLUSION

Under open multi-view learning contexts, characterized by ambiguity, we confront two principal challenges: Mitigating the adverse effects of variability and recognizing unknown classes with multi-view information. In this paper, the inherent ambiguity of these synthetic points is managed by creating samples located at taxon boundaries, using mixed samples from different taxa, and applying soft labels to manage the inherent ambiguity of these synthetic points. In this way, the generalization ability of the model is enhanced by adapting the model to the data distribution of potentially unknown categories in advance during the training phase. In addition, we design a one-step sparse inference model that proficiently eliminates superfluous information across multiple views. This is complemented by confidence score estimation, which gates view-specific representations to leverage prediction scores and enhance reliability. Additionally, the inter-class dispersion is adopted to ascertain the informativeness of the views, thereby amalgamating them into a cohesive final representation. The proposed AAML not only accurately classifies instances based on comprehensive multi-view data, but also extends its classification capabilities beyond the training data.

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
