# OpenReview forum: "Beyond the Known: Ambiguity-Aware Multi-view Learning"
_acmmm.org/ACMMM/2024/Conference — MM2024 Poster_

### Official Review · Reviewer_qHT5 · 2024-05-24

**Rating:** 3
**Confidence:** 4

**Summary:**

The paper introduces the Ambiguity-Aware Multi-view Learning Framework (AAML), designed to handle the inherent ambiguity in open multi-view learning environments. The framework integrates four synergistic modules to enhance generalizability and reliability by creating mixed samples with soft labels, implementing instance-level sparse inference, generating view-level confidence mappings, and employing centroid-guided adaptive fusion for consistent representation. This approach effectively prepares the model for unknown categories and improves classification accuracy and robustness. Extensive experiments validate the superior performance and stability of AAML compared to other state-of-the-art methods.

**Strengths:**

- The AAML integrates multiple synergistic modules to handle complexity in a systematic way, which is a strength in terms of providing a holistic solution. The centroid-guided adaptive fusion dynamically adjusts the contributions of different views, ensuring a cohesive and informative final representation.
- Extensive comparative experiments demonstrate AAML's superior performance and stability in open-world recognition scenarios.

**Limitations:**

1. The motivation behind the paper is not clearly articulated. The concept of ambiguity is not well-defined within the context of the paper. It is unclear whether ambiguity refers solely to unknown classes/outliers, or if it encompasses other aspects. This lack of clarity could make the title seem misleading or out of scope relative to the actual content and contributions of the work.

2. The paper claims that linear interpolation of two samples can produce outliers. However, linear interpolation should typically result in a combination of two existing images from the original distribution, not outliers. If this approach were to generate outliers, it would imply that existing mixup-based methods would fail, which contradicts current empirical evidence supporting their effectiveness.

3. How to combine Soft-Mixup Augmentation with the ground truth class TCP? How to obtain the predicted probability of the ground truth class TCP?

4. The paper lacks comprehensive comparisons with the methods like "Robust Multi-view Clustering with Incomplete Information" (TPAMI 2023) and "Dual Contrastive Prediction for Incomplete Multi-View Representation Learning" (TPAMI 2023).

5. The writing is not well-organized, and the annotations used in the equations are confusing. For example, the difference between $\mathbb{I} \tilde{y}$ and $\mathbb{I} \tilde{y}_i$ in Equations (4) and (5) is not explained. Additionally, the same notation $\lambda$ is used in both Equations (3) and (6) without clarification, potentially leading to misunderstandings. The meaning of $\Phi$ in Equation (9) is also not defined.

**Suitability:**

3

---

### Official Review · Reviewer_fRym · 2024-05-24

**Rating:** 4
**Confidence:** 2

**Summary:**

This paper addresses challenges in open multi-view learning, such as handling variability and recognizing unknown classes. By creating synthetic samples at taxon boundaries and using soft labels, the model's generalization to unknown categories is enhanced. The paper introduces a one-step sparse inference model that removes unnecessary multi-view information, complemented by a confidence scoring system to improve prediction reliability. Additionally, the method evaluates and integrates the informativeness of different views into a cohesive representation. The proposed framework not only classifies multi-view data accurately but is also adapted to recognize categories beyond the training data.

**Strengths:**

1. This paper is well written and easy to understand.
2. Good experimental results on four scenarios.

**Limitations:**

It is suggested that future revisions include a deeper exploration of the motivations behind specific design decisions, enhancing the reader's understanding of the reasons for these choices beyond just a description of the implemented actions.

**Suitability:**

2

---

### Official Review · Reviewer_5wDu · 2024-05-25

**Rating:** 4
**Confidence:** 4

**Summary:**

The paper introduces an innovative framework to address the challenge of ambiguity in open multi-view learning scenarios. The authors propose an Ambiguity-Aware Multi-view Learning Framework (AAML) that integrates four key modules: Soft-Mixup Augmentation, Instance-level Sparse Inference, View-level Confidence Network, and Centroid-Guided Adaptive Fusion. The framework aims to enhance the generalizability and reliability of predictors by adapting to the distribution of unknown samples and reconciling disparate views to distill essential patterns.

**Strengths:**

- The paper introduces an Ambiguity-Aware Multi-view Learning Framework (AAML) that addresses the challenge of ambiguity in open multi-view learning scenarios. This represents a novel approach to handling variability and unpredictability in real-world data.
- The framework is designed to adapt to the distribution of potentially unknown samples during training, which enhances its ability to generalize beyond the known categories and makes it robust for open-world recognition tasks.
- AAML specifically addresses the issue of ambiguity, which is often neglected in previous works but is crucial for real-world applications where models must operate under conditions that may differ from those seen during training.

**Limitations:**

- How does the concept of ambiguity in multi-view learning differ from uncertainty in single-view scenarios, and why is it crucial to address this?
- Can the authors elaborate on why mixed samples and soft labels are essential for adapting to the distribution of unknown samples? Meanwhile, I think that it may introduce more noise and uncertainty to the learning process.
- Are there existing methods or frameworks that the authors found inadequate in handling ambiguity in multi-view learning, and how does their work differ?
- Can the authors elaborate on the sparse inference process and how it contributes to filtering out redundant features? How is the sparsity enforced, and what is the role of the combined ℓ1-norm in this context?
- The paper introduces the concept of ambiguity in multi-view learning. I am confused that how is ambiguity specifically quantified and handled within the framework?
- What criteria were used to select the benchmark datasets (ESP-Game, Flower17, MNIST, Reuters, ORL, Youtube), and how do they represent the challenges of open multi-view learning?

- The paper mentions managing uncertainty with soft labels. How is uncertainty quantified in the experiments, and what is the impact on the decision-making process?

**Suitability:**

3

---

### Official Review · Reviewer_nzvx · 2024-05-28

**Rating:** 3
**Confidence:** 3

**Summary:**

This paper focuses on the practical problem of considerable ambiguity among open multi-view learning and propose a Ambiguity-Aware Multi-view Learning Framework (AAML), which consists of four key components that each geared towards achieving specific objectives to jointly address the unknown classes problem. Experimental results on six well-known multi-view datasets demonstrate that the effectiveness of AAML.

**Strengths:**

1. The studied problem is interesting.
2. This paper is well-written, well-organized, and easy to follow.

**Limitations:**

1. In the INTRODUCTION, the authors propose two questions. However, based on the narrative flow of the paper, the first question appears to be somewhat overlooked. It is suggested that the authors revisit the introduction and ensure the first question is adequately incorporated into the logical flow of the paper.
2. In the RELATED WORK section, some related works are missing: a) CL-MVSNet: Unsupervised Multi-view Stereo with
Dual-level Contrastive Learning, ICCV 2023; b) Self-Supervised Information Bottleneck for Deep Multi-View Subspace Clustering, TIP, 2023
3. The paper could be strengthened by providing a more detailed analysis of the effectiveness and limitations of the proposed approach. This factor could be significant for guiding future research and practical applications of the methodology.

**Suitability:**

2

---

### Meta-Review · Area_Chair_tJZw · 2024-07-01

**Recommendation:** Accept (Poster)
**Confidence:** 5

**Metareview:**

All reviewers lean towards acceptance of this paper. The AC concurs with the reviewers and recommends this paper for publication. The authors should address all the issues raised in the reviews in the final version of their paper.